# NAR-Former V2: Rethinking Transformer for Universal Neural Network Representation Learning

**Yun Yi**[1,3,†]    **Haokui Zhang**[2,3,*]    **Rong Xiao**[3]    **Nannan Wang**[1,*]    **Xiaoyu Wang**[3]
[1]Xidian University    [2]Northwestern Polytechnical University    [3]Intellifusion
yuny220@163.com    hkzhang1991@mail.nwpu.edu.cn
xiao.rong@intellif.com    nnwang@xidian.edu.cn    fanghuaxue@gmail.com

## Abstract

As more deep learning models are being applied in real-world applications, there is a growing need for modeling and learning the representations of neural networks themselves. An effective representation can be used to predict target attributes of networks without the need for actual training and deployment procedures, facilitating efficient network design and deployment. Recently, inspired by the success of Transformer, some Transformer-based representation learning frameworks have been proposed and achieved promising performance in handling cell-structured models. However, graph neural network (GNN) based approaches still dominate the field of learning representation for the entire network. In this paper, we revisit the Transformer and compare it with GNN to analyze their different architectural characteristics. We then propose a modified Transformer-based universal neural network representation learning model NAR-Former V2. It can learn efficient representations from both cell-structured networks and entire networks. Specifically, we first take the network as a graph and design a straightforward tokenizer to encode the network into a sequence. Then, we incorporate the inductive representation learning capability of GNN into Transformer, enabling Transformer to generalize better when encountering unseen architecture. Additionally, we introduce a series of simple yet effective modifications to enhance the ability of the Transformer in learning representation from graph structures. In encoding entire networks and then predicting the latency, our proposed method surpasses the GNN-based method NNLP by a significant margin on the NNLQP dataset. Furthermore, regarding accuracy prediction on the cell-structured NASBench101 and NASBench201 datasets, our method achieves highly comparable performance to other state-of-the-art methods. The code is available at https://github.com/yuny220/NAR-Former-V2.

## 1 Introduction

With the maturity of deep learning technology, an increasing number of deep neural network models of various sizes and structures are being proposed and implemented in academic research and industrial applications. In this process, the rapid deployment of networks and the design of new networks that meet task requirements are significant. To address this issue, researchers propose using machine learning models to solve the deployment and design problems of the models themselves. One popular strategy is encoding the input neural network and utilizing the resulting neural network representation to predict a specific target attribute directly without actually executing the evaluation program. In recent years, we have witnessed success in accelerating model deployment and design

---

[†]This work was done while Yun Yi was an intern at Intellifusion.
[*]Corresponding authors.

37th Conference on Neural Information Processing Systems (NeurIPS 2023).

processes with the help of neural network representations [47, 21, 3, 46, 26, 44, 12]. Taking the advantages of latency predictors [47, 21, 17, 55, 12, 2], significant time cost and expertise efforts can be saved by not having to carry out the time-consuming process of compilation, deployment, inference, and latency evaluation when engineers choose networks for application. Through the use of accuracy predictors [47, 26, 46, 4, 12, 44, 20], researchers can avoid the resource-intensive process of network training and instead perform a forward inference process to evaluate the accuracy of a multitude of networks. This measure dramatically reduces the time cost associated with network design.

Although the vanilla Transformer is designed for natural language processing, Transformer architecture has found widespread adoption across diverse fields owing to its strengths in global modeling and parallelizable computation [24, 22, 23, 14, 41, 8]. Very recently, several researchers have attempted to learn appropriate representations for neural networks via Transformer [47, 26]. These methods have indeed achieved leading performance on relevant tasks. Nevertheless, they are mainly designed for encoding the architecture of cells (basic micro units of repeatable neural networks) in cell-structured networks. As shown in the latency prediction experiment in NAR-Former (Transformer-based neural architecture representation learning framework) [47], poor generalization performance occurs when the depth of the input architecture reaches hundreds of layers. In the development process of neural network representation learning, Graph neural network (GNN) [16, 35] is also a promising technique for learning neural network representations [21, 37, 19, 43, 12]. They model the input neural architecture as a directed acyclic graph (DAG) and operate on the graph-structured data, which comprises the node information matrix and adjacency matrix. Recently, the NNLP [21] introduced a dedicated latency prediction model based on GNNs, which is capable of encoding the complete neural network having hundreds of layers and achieves a cutting-edge advance.

In fact, both cell-structured architectures and complete neural networks are widely used in various applications. Cell-structured models offer good scalability, allowing for easy scaling by adding or removing cells. This adaptability makes them suitable for addressing problems of different complexities and data sizes, while also facilitating incremental model development and deployment. Complete neural networks provide better flexibility in connectivity and can achieve higher accuracy in certain cases. Furthermore, in some cases, such as in latency estimation, encoding the complete network is necessary. To handle various network architectures in different tasks, both GNN-based and Transformer-based models are necessary. However, this issue of utilizing multiple architectures can introduce constraints that may not be conducive to practical applications. For instance, when a designed network requires specific attributes, having similar model structures and high-accuracy predictions for different attributes can reduce code redundancy and improve work efficiency.

In this paper, we build upon the research conducted in NAR-Former [47] and present a novel framework called NAR-Former V2 for universal neural network representation learning. Our framework can handle cell-structured networks as well as learn representations for the entire network. To accomplish this, we incorporate graph-specific properties into the vanilla Transformer and introduce a graph-aided attention-based Transformer block. This approach combines the strengths of both Transformer and graph neural networks (GNNs). Extensive experiments are conducted to evaluate our proposed framework. Results show that:(1) our method can be applied to predict different attributes, can outperform the state-of-the-art method in latency prediction on the NNLQP dataset [21], and can achieve promising results in accuracy prediction on the NAS-Bench-101 and NAS-Bench-201 datasets [48, 10]; (2) our method has good scalability, which is capable of encoding network having only a few operations or complete neural networks that have hundreds of operations.

## 2   Related work

### 2.1   Representation and attribute prediction of neural networks

Neural network representation learning is the base for evaluating the attributes of different networks via machine learning models. Early methods [7, 20] construct representation models for learning neural network representation based on LSTM and MLP. Peephole [7] inputs the embedding of each layer to LSTM to predict accuracy, which neglects the topological structure and is limited to handling only sequential architectures. Later, in order to better capture the structural information of the network, an accuracy predictor [44] uses a binary path encoding with a length equal to the number of possible paths from input to output given in terms of operations, where the element at the corre-

sponding position of the path presenting in the input network is set to 1. When the neural network is regarded as a directed acyclic graph, the adjacency matrix describes the connection between nodes, so it is naturally used to encode the topological structure of the neural network. NAS-Bench-101 [48] proposed to encode the given neural network as a concatenated vector of a flat adjacency matrix and a list of node labels. Many other methods [21, 17, 3, 12, 43, 19] realize accuracy and latency prediction by directly inputting the original two-dimensional adjacency matrix together with the node information matrix to GNN, which can realize the explicit encoding of the input network topology. There also are some works exploring other ways of obtaining representations of neural networks, such as by mimicking actual data processing [31] or message exchange [50] of neural networks, and by deriving a number of persistent topology measures for DNNs [5]. Recently, other methods have focused on enhancing the original GNN [4] or introducing transformers [26, 47] to obtain more meaningful neural network representations.

## 2.2  Transformer

Transformer [41] is a self-attention-based neural network architecture that has revolutionized natural language processing [28, 8, 33] and has been adopted in many other fields [54, 24, 22, 23, 47, 26, 11, 14]. Because of its excellent modeling capability and wide range of applications, there is a growing interest in the direction of further enhancing its performance [53, 1, 39, 6], improving computing efficiency [23, 40, 34], combining with other backbone networks [15, 18, 38, 29, 13], and even taking inspiration from it to optimize other types of network structures, such as ConvNets [45, 51, 25, 9, 52]. Transformer has recently been successfully introduced into neural network representation learning [26, 47]. TNASP [26] inputs the sum of the operation type embedding matrix and Laplacian matrix into the standard Transformer. NAR-Former [47], on the other hand, encodes each operation and connection information of this operation into a token and inputs all tokens into a proposed multi-stage fusion transformer. Excellent attribute prediction results have been achieved on cell-based datasets by using these methods. However, the strong long-range modeling ability of the self-attention mechanism may also result in subtle local variation affecting the representation of all tokens. Due to the potential impact of this feature on the generalization ability, although NAR-Former [47] has made attempts to encode complete neural networks, the results are still unsatisfactory.

## 2.3  Graph neural network

GNNs are designed to handle graph-structured data, which is a fundamental representation for many real-world problems such as social network analysis and recommendation systems [49, 32]. Given that neural networks can be viewed as graphs, GNN-based models have emerged as a prominent and widely adopted approach for neural network representation learning [21, 17, 3, 12, 43, 19]. GNNs show generalization ability through a simple mechanism of aggregating information from neighbors. For instance, the recently proposed GNN-based model [21] can obtain representations of neural networks with hundreds of layers and achieves new state-of-the-art results in latency prediction, even if the input network structure has not been seen during training. Nevertheless, the simple structural characteristics of GNNs, which contribute to their strong generalization ability, also lead to the need for further improvement in the performance of methods based on original GNN in cellular structure and complete neural network representation learning. Therefore, it is a promising approach for neural network representation learning to combine the Transformer and GNN to leverage the strengths of both models.

## 3  Method

Our final framework diagram for realizing neural architecture representation learning and attribute prediction is shown in Fig. 1, consisting of three main phases: neural network encoding, backbone-based representation learning, and attribute predicting using prediction heads. We will introduce the motivation and details of the backbone network design in Sec. 3.1 and Sec. 3.2, and the details of the entire model in Sec. 3.3.

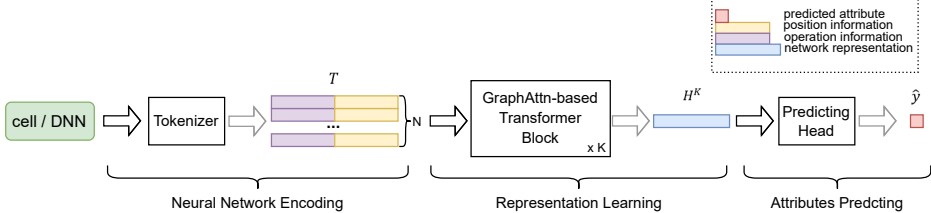

Figure 1: Overview of attribute prediction model.

## 3.1 Motivation

As mentioned in the Sec. 1, Transformer-based models have demonstrated remarkable performance in encoding and learning representations of neural networks when the input is in the form of cells. However, when dealing with complete deep neural networks (DNNs) consisting of hundreds of layers, or when the depth of the input data is unseen during training, they may sometimes exhibit poorer performance compared to GNN-based methods. Additionally, as highlighted in [21], real-world applications often show significant differences in the topologies and depths between training and test samples. Consequently, representation learning models must possess strong generalization abilities for handling unseen data. In this regard, GNN-based models appear to achieve better performance.

This observation has prompted us to reconsider the two types of inputs, namely cells and complete DNNs, as well as the two representation learning models, the Transformer and GNN. Through a detailed comparative analysis of the structures of the Transformer and GNN, we speculate that the insufficient generalization capability of Transformer-based methods may be attributed to its structure and computation characteristics. As we know, the self-attention structure in transformers is a crucial design that allows for the effective extraction of global features in a data-driven manner. However, this structure becomes a double-edged sword when learning network representations. For input neural networks with depths of hundreds of layers, the Transformer's impressive capability to capture global information can sometimes lead to excessive sensitivity. This stems from the fact that the Transformer models interactions between all tokens using its self-attention mechanism, treating the entire sequence as a fully connected graph. This dense attention mechanism can give rise to a particular issue: even a subtle variation, such as reducing the kernel size in a layer from $5 \times 5$ to $3 \times 3$, can affect the representation of all other layers, ultimately leading to significant differences in the final representation. As a result of this issue, the trained model may be biased toward fitting the training data. Consequently, when the model is employed for inferring architectures outside the training data distribution, it yields inferior results and demonstrates poorer generalization performance. The corresponding experiments are presented in Sec. 4.4.

## 3.2 Transformer grafted with GNN

Fig. 2 shows the vanilla transformer block, GNN block, and our proposed graph-aided attention Transformer block. As shown in Fig.1 (a), the vanilla Transformer block has two major parts:

$$\hat{H}^l = \text{SelfAttn}(\text{LN}(H^{l-1})) + H^{l-1}, \tag{1}$$

$$H^l = \text{FFN}(\text{LN}(\hat{H}^l)) + \hat{H}^l, \tag{2}$$

where $H^l$ is the feature for the layer $l$. $\hat{H}^l$ is an intermediate result. SelfAttn, FFN, and LN refer to self-attention, feed-forward network, and layer normalization, respectively. GNN block just has one major part, where the representation is updated following:

$$\hat{H}^l = \text{GraphAggre}(H^{l-1}, A) + W_r^l H^{l-1}, \tag{3}$$

$$H^l = \text{L}_2(\hat{H}^l), \tag{4}$$

where $\text{GraphAggre}(H^{l-1}, A) = W_a^l(\text{Norm}(A)H^{l-1})$. $A \in \mathbb{R}^{N \times N}$ is the adjacency matrix, and $\text{L}_2$ denotes the l2-normalization function. $W$ with different superscripts and subscripts represents different learnable transformation matrices.

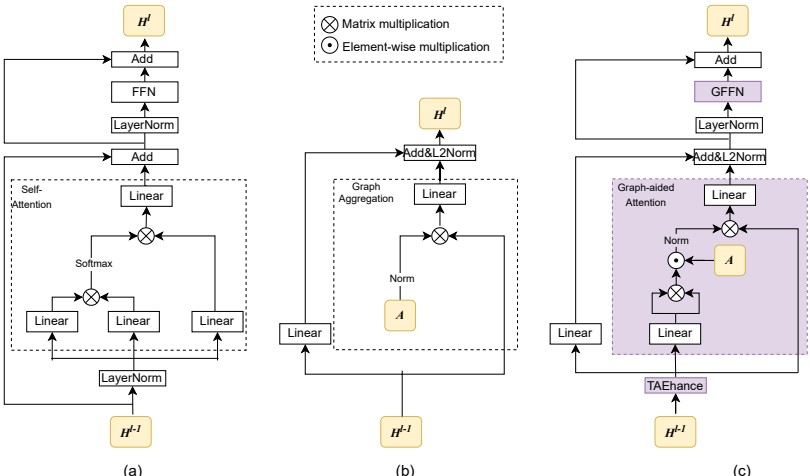

Figure 2: Diagrams of three modules. (a) The vanilla Transformer block [42]. (b) The GNN layer with mean aggregator [16]. (c) The proposed graph-aided attention Transformer block.

Comparing formulas (3) and (4) with formulas (1) and (2), we can observe two major differences between the Transformer block and GNN block:

- The Transformer utilizes self-attention to fuse information from a global perspective, while the GNN uses graph aggregation to fuse neighbor information based on the adjacency matrix.

- The Transformer block includes an additional FFN (Feed-Forward Network) component, which enhances information interaction between channels.

The advantage of self-attention lies in its data-driven structure, allowing for flexible adjustment of information fusion weights based on the input data. On the other hand, the advantage of GNN is that graph aggregation focuses on the topological structure. These two advantages are not contradictory to each other. Consequently, we have naturally come up with an idea to combine the strengths of self-attention and graph aggregation. This approach inherits the flexibility of self-attention while benefiting from the good generalization capability of graph aggregation.

To implement this idea, we consider the neural network encoded as a graph, with the operations or layers in the network treated as nodes. Assuming the graph has $N$ nodes, the transformer layer we have designed for universal neural network representation learning (Fig. 2 (c)) is calculated as follows:

$$\widetilde{H}^l = \text{TAEnhance}(H^{l-1}, D), \tag{5}$$

$$\hat{H}^l = \text{L}_2(\text{GraphAttn}(\widetilde{H}^l, A) + W_r^l \widetilde{H}^l), \tag{6}$$

$$H^l = \text{GFFN}(\text{LN}(\hat{H}^l)) + \hat{H}^l, \tag{7}$$

where $D \in \mathbb{R}^{N \times 1}$ is a vector that records the number of nodes directly connected to each node. The Graph-aided Attention (GraphAttn) module is responsible for performing attention calculations using the properties of the graph structure to adjust global self-attention. The Type-aware Enhancement module (TAEnhance) is utilized to further enhance the representation. We introduce a Grouped Feed-Forward Network (GFFN) by introducing group linear transformation into the original FFN. In the following sections, we will provide a detailed introduction to each component.

**Graph-aided attention**   In the proposed graph-aided attention, we employ the adjacency matrix to govern the attention calculation range. Moreover, the adjacency matrix characterizes the inter-layer connection relationships within the neural network, enabling the model to acquire topology knowledge. Hence, we define this module as the Graph-aided Attention module:

$$X^l = \text{Sigmoid}(W_q^l \widetilde{H}^l + b_q^l), \tag{8}$$

$$S^l = (X^l X^{lT}/\sqrt{d}) \odot A, \tag{9}$$

$$Z^l = W_a^l(\text{Norm}(S^l)\widetilde{H}^l) + b_a^l. \tag{10}$$

The $\text{Norm}(\cdot)$ means that for each node, the attention weights between it and other nodes are transformed to $(0, 1)$ by dividing it by the sum. The character $b$ with different superscripts and subscripts represents different learnable biases. The $d$ refers to the feature dimension of $X^l$. Note that simply using the adjacency matrix to control the attention map in self-attention is insufficient. To make this approach work, we have discarded the original softmax operation in self-attention and replaced it with a linear attention mechanism. This is because the softmax operation tends to focus excessively on the current node while neglecting neighboring nodes. Consequently, we have inserted a sigmoid activation function before the linear attention to ensure that all values in the attention map are positive. For further comparisons with the original self-attention, please refer to the supplementary.

**Type-Aware enhancement module**  The connection between a layer and other layers is related to the type of that layer. Therefore, the number of connected layers in each layer can be used to assist the model in learning the type of layer. By fully utilizing the internal characteristics of this graph-structured data, it is beneficial to improve the learned representations. The enhanced representation is obtained by:

$$\text{TAEnhance}(H^{l-1}, D) = \text{Sigmoid}(W_d^l D + b_d^l) \odot H^{l-1}. \tag{11}$$

### 3.3 Universal representation learning for neural network

In this subsection, we will present the construction of a comprehensive framework for neural network encoding and representation learning based on the proposed enhanced Transformer block. We will also explain how this framework can be utilized to predict network attributes. The overall system, as depicted in Fig. 1, is composed of three consecutive stages: neural network encoding, representation learning, and attribute prediction.

**Neural network encoding**  We have taken inspiration from the tokenizer used in NAR-Former [47] and made certain modifications to encode the input neural network. For a given network consisting of $N$ layers or operations, whether it is a cell architecture or a complete DNN, we represent it as a sequence feature comprising vectors corresponding to each layer: $T = (t_1, t_2, \cdots, t_N) \in \mathbb{R}^{N \times C}$. In NAR-Former, for each vector of the current node, it encapsulates information of operation, self-position, and the position of its source nodes: $t_i = (t_i^{\text{op}}, t_i^{\text{self}}, t_i^{\text{sour}}) \in \mathbb{R}^C$.

Following the encoding scheme proposed in NAR-Former [47], we use the position encoding formula [41, 30] to transform the single real-valued numbers (e.g. operation labels and node position indices) of relevant information into a higher-dimensional space. We denote this mapping scheme as $f_{\text{PE}}(\cdot)$.

For the operation encoding $t_i^{\text{op}}$, there are slight differences in the specific encoding content for input networks of different scales. If the input is in the form of a cell architecture [48, 10], there are usually no more than ten different options for each architecture operation. In this case, we directly assign category labels to all candidate operations, and then use the function $f_{\text{PE}}(\cdot)$ to encode the label of the operation to obtain $t_i^{\text{op}}$. However, when a complete DNN is used as input [21], more abundant operational information can be extracted and encoded. In this case, we first use one-hot vectors, which ensure the same distance between different categories, to encode the type of operation (e.g. convolution, batch normalization, ReLU, concatenation). Then use the function $f_{\text{PE}}(\cdot)$ to encode the properties (e.g. kernel size, number of groups) of the operation, which is then concatenated with the one-hot type vector as $t_i^{\text{op}}$. NAR-Former [47] uses the concatenation of $t_i^{\text{self}}$ and $t_i^{\text{sour}}$ to encode information flow. Since our improved transformer can obtain the topology information of the network with the help of the adjacency matrix, these two parts of the encoding are not necessarily needed in our method. Details about the encoding in the different experiments are provided in our released code and supplementary material.

**Representation learning**   The model for learning neural network representations $H^K$ is constructed by stacking multiple instances of our proposed enhanced Transformer blocks. These improvements, specifically tailored for neural network data, allow the model to learn representations that are more meaningful and exhibit enhanced generalization capabilities.

**Attributes predicting**   Taking the representation $H^K$ as input, the target attribute can be predicted by using the predicting head:

$$\hat{y} = -\text{logsigmoid}(\text{FC}(\text{ReLU}(\text{FC}(\text{ReLU}(\text{FC}(H^K)))))). \tag{12}$$

Currently, among the various attributes of the network, accuracy and latency are the two main types of predicted objects. Because they have extremely high acquisition costs and are the primary manifestation of network performance and efficiency.

For latency prediction, due to the strong correlation between batch size, memory access, parameter quantity, and FLOPs with network latency, the encoding corresponding to these characteristics and the representation $H^K$ are input into the predicting head together. To train the whole latency prediction model, the mean square error (MSE) function is adopted to measure the difference between predicted results and the ground truths.

For accuracy prediction, in addition to the MSE loss function, architecture consistency loss (AC_loss) and sequence ranking related loss (SR_loss) proposed by NAR-Former [47] are also used. Following NAR-Former, we employed a hierarchical fusion strategy in accuracy prediction experiments. We use a simplified approach, which computes the weighted sum of the outputs of each transformer layer with adaptive weights.

## 4   Experiments

In this section, we conduct experiments on NNLQP [21], NAS-Bench-101 [48], and NAS-Bench-201 [10] to evaluate the performance of our NAR-Former V2. A series of ablation experiments were performed to corroborate the effectiveness of our design details. More experiments and more details about implementation will be provided in the supplementary materials.

### 4.1   Implementation details

**Model details**   For latency experiments, the number of GraphAttn-based Transformer blocks is set to 2, which is the same as the baseline [21]. As for accuracy predicting, we fix the number of Transformer blocks to 6 to align with the standard Transformer used in the baseline [47].

**Training details**   All experiments were trained using the Adam optimizer. We used a linear learning rate decay strategy with a warm-up, in which the learning rate uniformly increased to 0.001 (latency prediction) or 0.0001 (accuracy prediction) during the first 10% of the training steps and then gradually decayed to 0. The batch size was fixed at 16. Our models are trained on a machine with a GeForce RTX 3090 GPU. To avoid randomness, each model was trained 12 times and the two experiments with the best and worst indicators were discarded.

### 4.2   Latency prediction

We conduct latency prediction on the recently released NNLQP dataset [21], which comprises 20000 complete deep learning networks and their corresponding latencies on the target hardware. This dataset has 10 different types of networks (referring to the first column of Tab. 1), with 2000 networks per type. Following NNLP [21], we use Mean Absolute Percentage Error (MAPE) and Error Bound Accuracy (Acc($\delta$)) to measure the deviations between latency predictions and ground truths. The lower the MAPE, the higher the prediction accuracy, while the opposite is true for Acc($\delta$).

Here, we considered two different scenarios. In the first scenario, the training and testing sets are from the same distribution. We constructed the training set with the first 1800 samples from each of the ten network types, and the remaining 2000 networks were used as the testing set. The detailed results are shown in Tab. 1. When testing with all test samples, the average MAPE of our method is 0.4% lower than that of NNLP [21], and the average Acc(10%) is 1.16% higher than that of NNLP.

Table 1: Latency prediction on NNLQP [21]. Training and test sets have the same distribution.

| Test Model | NAR-Former [47] | MAPE↓ NNLP [21] avg / best | Ours avg / best | NAR-Former [47] | Acc(10%)↑ NNLP [21] avg / best | Ours avg / best |
|---|---|---|---|---|---|---|
| All | 22.37% | 3.47% / 3.44% | 3.07% / 3.00% | 35.00% | 95.25% / 95.50% | 96.41% / 96.30% |
| AlexNet | 26.25% | 6.37% / 6.21% | 6.18% / 5.97% | 27.00% | 81.75% / 84.50% | 81.90% / 84.00% |
| EfficientNet | 13.91% | 3.04% / 2.82% | 2.34% / 2.22% | 45.50% | 98.00% / 97.00% | 98.50% / 100.0% |
| GoogleNet | 16.00% | 4.18% / 4.12% | 3.63% / 3.46% | 39.00% | 93.70% / 93.50% | 95.95% / 95.50% |
| MnasNet | 15.76% | 2.60% / 2.46% | 1.80% / 1.70% | 33.00% | 97.70% / 98.50% | 99.70% / 100.0% |
| MobileNetV2 | 15.19% | 2.47% / 2.37% | 1.83% / 1.72% | 39.00% | 99.30% / 99.50% | 99.90% / 100.0% |
| MobileNetV3 | 16.88% | 3.50% / 3.43% | 3.12% / 2.98% | 36.00% | 95.35% / 96.00% | 96.75% / 98.00% |
| NasBench201 | 43.53% | 1.46% / 1.31% | 1.82% / 1.18% | 55.50% | 100.0% / 100.0% | 100.0% / 100.0% |
| SqueezeNet | 24.33% | 4.03% / 3.97% | 3.54% / 3.34% | 23.00% | 93.25% / 93.00% | 95.95% / 96.50% |
| VGG | 23.64% | 3.73% / 3.63% | 3.51% / 3.29% | 26.50% | 95.25% / 96.50% | 95.85% / 96.00% |
| ResNet | 28.18% | 3.34% / 3.25% | 3.11% / 2.89% | 25.50% | 98.40% / 98.50% | 98.55% / 99.00% |

Table 2: Latency prediction on NNLQP [21]. "Test Model = AlexNet" means that only AlexNet models are used for testing, and the data from the other 9 model families are used for training. The best results refer to the lowest MAPE and corresponding ACC (10%) in 10 independent experiments. *: obtained based on the released code without using its fine-tuning step.

| Metric | Test Model | FLOPs | FLOPs +MAC | nn-Meter [55] | TPU [17] | BRP-NAS [12] | NAR-Former [47]* | NNLP [21] (avg / best) | Ours (avg / best) |
|---|---|---|---|---|---|---|---|---|---|
| MAPE↓ | AlexNet | 44.65% | 15.45% | 7.20% | 10.55% | 31.68% | 46.28% | 10.64% / 9.71% | 24.28% / 18.29% |
| | EfficientNet | 58.36% | 53.96% | 18.93% | 16.74% | 51.97% | 29.34% | 21.46% / 18.72% | 13.20% / 11.37% |
| | GoogleNet | 30.76% | 32.54% | 11.71% | 8.10% | 25.48% | 24.71% | 13.28% / 10.90% | 6.61% / 6.15% |
| | MnasNet | 40.31% | 35.96% | 10.69% | 11.61% | 17.26% | 26.70% | 12.07% / 10.86% | 7.16% / 5.93% |
| | MobileNetV2 | 37.42% | 35.27% | 6.43% | 12.68% | 20.42% | 25.74% | 8.87% / 7.34% | 6.73% / 5.65% |
| | MobileNetV3 | 64.64% | 57.13% | 35.27% | 9.97% | 58.13% | 33.99% | 14.57% / 13.17% | 9.06% / 8.72% |
| | NasBench201 | 80.41% | 33.52% | 9.57% | 58.94% | 13.28% | 105.71% | 9.60% / 8.19% | 9.21% / 7.89% |
| | ResNet | 21.18% | 18.91% | 15.58% | 20.05% | 15.84% | 40.37% | 7.54% / 7.12% | 6.80% / 6.44% |
| | SqueezeNet | 29.89% | 23.19% | 18.69% | 24.60% | 42.55% | 74.59% | 9.84% / 9.52% | 7.08% / 6.56% |
| | VGG | 69.34% | 66.63% | 19.47% | 38.73% | 30.95% | 44.26% | 7.60% / 7.17% | 15.40% / 14.26% |
| | Average | 47.70% | 37.26% | 15.35% | 21.20% | 30.76% | 45.17% | 11.55% / 10.27% | 10.55% / 9.13% |
| Acc(10%)↑ | AlexNet | 6.55% | 40.50% | 75.45% | 57.10% | 15.20% | 7.60% | 59.07% / 64.40% | 24.65% / 28.60% |
| | EfficientNet | 0.05% | 0.05% | 23.40% | 17.00% | 0.10% | 15.15% | 25.37% / 28.80% | 44.01% / 50.20% |
| | GoogleNet | 12.75% | 9.80% | 47.40% | 69.00% | 12.55% | 24.35% | 36.30% / 48.75% | 80.10% / 83.35% |
| | MnasNet | 6.20% | 9.80% | 60.95% | 44.65% | 34.30% | 20.90% | 55.89% / 61.25% | 73.46% / 81.60% |
| | MobileNetV2 | 6.90% | 8.05% | 80.75% | 33.95% | 29.05% | 20.70% | 63.03% / 72.50% | 78.45% / 83.80% |
| | MobileNetV3 | 0.05% | 0.05% | 23.45% | 64.25% | 13.85% | 16.05% | 43.26% / 49.65% | 68.43% / 70.50% |
| | NasBench201 | 0.00% | 10.55% | 60.65% | 2.50% | 43.45% | 0.00% | 60.70% / 70.60% | 63.13% / 71.70% |
| | ResNet | 26.50% | 29.80% | 39.45% | 27.30% | 39.80% | 13.25% | 72.88% / 76.40% | 77.24% / 79.70% |
| | SqueezeNet | 16.10% | 21.35% | 36.20% | 25.65% | 11.85% | 11.40% | 58.69% / 60.40% | 75.01% / 79.25% |
| | VGG | 4.80% | 2.10% | 26.50% | 2.60% | 13.20% | 11.45% | 71.04% / 73.75% | 45.21% / 45.30% |
| | Average | 7.99% | 13.20% | 47.42% | 34.40% | 21.34% | 14.09% | 54.62% / 60.65% | 62.70% / 67.40% |

When tested on various types of network data separately, except for the NASBench201 family, our method consistently outperforms NNLP. This indicates that our improved transformer, which utilizes the structural characteristics of the graph, has learned more reasonable representations than the original GNN. Comparison with NAR-Former's results also proves that our NAR-Former V2 is more effective than the original transformer-based model in learning representations of complete DNNs.

The second scenario has more practical application significance, that is, the network type needed to be inferred is not seen during the training process. There are ten sets of experiments in this part, with each set taking one type of network as the test set, while all samples from the other nine types of networks are used as the training set. As shown in Tab. 2, it can be seen that using only FLOPs and memory access information to predict latency is not enough. Suffering from the gap between the accumulation of kernel delays and the actual latency, kernel-based methods (TPU[17] and nn-Meter[55]) perform worse than the GNN-based model NNLP that directly encodes and predicts the entire network. Despite encoding the entire network directly, because of the sensitive nature introduced by the global modeling computational properties of the transformer, it may result in the transformer-based NAR-Former being less able to generalize to networks that have not been seen at the training stage. Benefiting from considering the entire input network and grafting GNN into the transformer, our method achieves the best MAPE and Acc(10%) on the average indicators of 10

experimental groups. Compared with the second-best method NNLP, the average Acc(10%) of our method has a marked increase of 8.08%.

### 4.3 Accuracy prediction

#### 4.3.1 Experiments on NAS-Bench-101

NAS-Bench-101 [48] provides 423624 different cell architectures and the accuracies of the complete neural network constructed based on each cell on different datasets. Following [26], 0.1% and 1% of the whole data are used as the training set, and another 200 samples are used for validation. We use Kendall's Tau [36] to evaluate the correlation between the predicted sequence and the real sequence, and a higher value indicates better results.

Kendall's Tau is calculated on the whole dataset or 100 testing samples. We report the average results of our predictor in 10 repeated experiments. **Results** are shown in Tab. 3. When only

Table 3: Accuracy prediction on NAS-Bench-101 [48]. "SE" denotes the self-evolution strategy proposed by TNASP [26].

| Backbone | Method | Training Samples | | |
|---|---|---|---|---|
| | | 0.1% (424) | 0.1% (424) | 1% (4236) |
| | | Test Samples | | |
| | | 100 | all | all |
| CNN | ReNAS [46] | 0.634 | 0.657 | 0.816 |
| LSTM | NAO [27] | 0.704 | 0.666 | 0.775 |
| | NAO+SE | 0.732 | 0.680 | 0.787 |
| GNN | NP [43] | 0.710 | 0.679 | 0.769 |
| | NP + SE | 0.713 | 0.684 | 0.773 |
| | CTNAS [3] | 0.751 | - | - |
| Transformer | TNASP [26] | 0.752 | 0.705 | 0.820 |
| | TNASP + SE | 0.754 | 0.722 | 0.820 |
| | NAR-Former [47] | 0.801 | 0.765 | **0.871** |
| | NAR-Former V2 | **0.802** | **0.773** | 0.861 |

424 samples were available for training, our method achieved the highest Kendall's Tau. We achieve 0.773 when tested using the whole testing set, which is 0.8% and 8.9% higher than the transformer-based model [47] and GNN-based model [43], respectively. This proves that the modifications we made to the transformer based on inspiration from GNN are effective.

#### 4.3.2 Experiments on NAS-Bench-201

NAS-Bench-201 [10] is another cell-based dataset, which contains 15625 cell-accuracy pairs. Following [26], 5% and 10% of the whole data is used as the training set and another 200 samples are used for validation.

We use Kendall's Tau [36] computed on the whole dataset as the evaluation metric in this part. The average results of our predictor of 10 runs are reported. **Results** are shown in Tab. 4. The conclusion of this experiment is similar to Sec. 4.3.1. When compared with the second-best method, a substantial improvement (2.5%) of Kendall's Tau can be seen in the setting of

Table 4: Accuracy prediction on NAS-Bench-201 [10]. "SE" denotes the self-evolution strategy proposed by TNASP [26].

| Backbone | Model | Training Samples | |
|---|---|---|---|
| | | (781) | (1563) |
| | | 5% | 10% |
| LSTM | NAO [27] | 0.522 | 0.526 |
| | NAO + SE | 0.529 | 0.528 |
| GNN | NP [43] | 0.634 | 0.646 |
| | NP + SE | 0.652 | 0.649 |
| Transformer | TNASP [26] | 0.689 | 0.724 |
| | TNASP + SE | 0.690 | 0.726 |
| | NAR-Former [47] | 0.849 | **0.901** |
| | NAR-Former V2 | **0.874** | 0.888 |

training with 781 samples. A more comprehensive comparison with the baseline [47] using additional metrics can be found in the supplementary materials.

*Regarding the average Kendall's Tau under different training settings, our method is on par with NAR-Former with 0.812 on nas-bench-101 and slightly higher than NAR-former by 0.6% on NAS-Bench-201. Therefore, compared to NAR-Former, NAR-Former V2 achieves comparable accuracy prediction performance. In latency prediction experiments, our proposed model exhibits a clear advantage over NNLP and outperforms NAR-Former by a significant margin. In summary, by incorporating the strengths of GNN, the universal representation learning framework NAR-Former V2 is significantly enhanced. NAR-Former V2 addresses the shortcomings of NAR-Former, which was overly sensitive when handling complete network structures, while still retaining the outstanding performance of NAR-Former when handling cell-structured networks.*

### 4.4 Ablation studies

In this section, we conducted a series of ablation experiments on the NNLQP dataset to investigate the impact of various modifications. The results from Rows (2) and (3) in Table 5 indicate that for type encoding without numerical relationships, using one-hot vectors with equidistant proper-

Table 5: Ablation studies on NNLQP [21]. "PE" denotes position encoding.

| Row | Structure | Op Type | Op Attributes | Graph-Attn | GFFN | TA-Enhance | MAPE↓ | Acc(10%)↑ | Acc(5%)↑ |
|---|---|---|---|---|---|---|---|---|---|
| 1(Baseline) | GNN | One-hot | Real Num | - | - | - | 3.48 | 95.26 | 77.80 |
| 2 | GNN | PE | PE | - | - | - | 3.43(-0.05) | 95.11(-0.15) | 79.58(+1.78) |
| 3 | GNN | One-hot | PE | - | - | - | 3.33(-0.15) | 95.57(+0.31) | 80.19(+2.39) |
| 4 | Transformer | One-hot | PE | ✓ | - | - | 3.20(-0.28) | 96.00(+0.74) | 81.86(+4.06) |
| 5 | Transformer | One-hot | PE | ✓ | ✓ | - | 3.20(-0.28) | 96.06(+0.80) | 81.76(+3.96) |
| 6 | Transformer | One-hot | PE | ✓ | ✓ | ✓ | 3.07(-0.41) | 96.41(+1.15) | 82.71(+4.91) |

ties across different categories is more suitable. Comparing Row (3) in Table 5 with Row (4), we observe that introducing GNN characteristics into the Transformer improves the model's ability to learn effective representations and achieve more accurate predictions compared to using the original GNN. When replacing the FFN with the GFFN module with eight groups (Row (5)), the number of model parameters reduces to approximately one-eighth of that in Row (4), without a significant decrease in prediction accuracy. Compared to Row (5), Row (6) demonstrates an increase of 0.35% in ACC(10%) and 0.95% in ACC(5%). This confirms the role of the type-aware enhancement module in further refining and enhancing the rationality of the representations.

To verify our hypothesis regarding the generalization ability of the network and the effectiveness of the proposed graph-aided attention, we conducted comparative experiments in scenarios where the training and testing data have different distributions. The results of these ex-

Table 6: The influence of using different attentions. Test on EfficientNet family.

| Attention | MAPE↓ | ACC(10%)↑ |
|---|---|---|
| Global | 16.88% | 36.32% |
| Local | 13.20% | 44.01% |

periments are presented in Table 6. In order to perform the experiment on global attention, we excluded the step of multiplying the adjacency matrix $A$ in Equation 9, and instead replaced $S^l$ with $X^l X^{lT}/\sqrt{d}$. Results in Table 6 demonstrate that incorporating the adjacency matrix to restrict the scope of attention calculation is indeed beneficial for latency prediction on unseen data. The model utilizing graph-aided attention exhibited a significant improvement of 7.68% in ACC(10%) compared to the model using global attention.

## 5   Conclusion

In this paper, we combine the strengths of Transformer and GNN to develop a universal neural network representation learning model. This model is capable of effectively processing models of varying scales, ranging from several layers to hundreds of layers. Our proposed model addresses the limitations of previous Transformer-based methods, which exhibited excessive sensitivity when dealing with complete network structures. However, it still maintains exceptional performance when handling cell-structured networks. In future work, we will focus on optimizing the design of the representation learning framework and applying it to a broader range of practical applications. Such as using the proposed model to search for the best mixed precision model inference strategies.

## Acknowledgement

This work was supported in part by the National Key Research and Development Program of China under Grant 2018AAA0103202; in part by the National Natural Science Foundation of China under Grants U22A2096 and 62036007; in part by the Technology Innovation Leading Program of Shaanxi under Grant 2022QFY01-15; in part by Open Research Projects of Zhejiang Lab under Grant 2021KG0AB01 and in part by the Fundamental Research Funds for the Central Universities under Grant QTZX23042.

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
