# Supplementary Materials for NAR-Former V2: Rethinking Transformer for Universal Neural Network Representation Learning

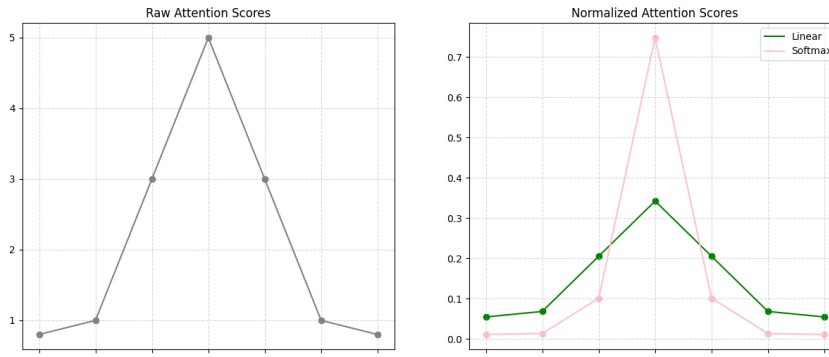

Figure 1: Left: Raw attention scores. Right: Normalized attention scores processed by two different normalization methods.

## A More detailed comparison with original self-attention

As introduced in the main text, linear graph-aided attention is defined as:

$$X^l = \text{Sigmoid}(W_q^l \widetilde{H}^l + b_q^l), \tag{1}$$

$$S^l = (X^l X^{lT}/\sqrt{d}) \odot A, \tag{2}$$

$$Z^l = W_a^l(\text{Norm}(S^l)\widetilde{H}^l) + b_a^l. \tag{3}$$

Compared with the self-attention in vanilla Transformer:

$$Q^l = W_q^l H^{l-1} + b_q^l, K^l = W_k^l H^{l-1} + b_k^l, V^l = W_v^l H^{l-1} + b_v^l, \tag{4}$$

$$S^l = \text{Softmax}(Q^l K^{lT}/\sqrt{d}), \tag{5}$$

$$Z^l = W_a^l(S^l V) + b_a^l, \tag{6}$$

in addition to the control of attention calculation range detailed in the main text, the main differences between the two attention schemes lie in:

- the graph-aided attention doesn't use the softmax function;
- the graph-aided attention adopts sigmoid activation function.

As shown in the Fig. 1, the softmax function helps the self-attention layer in directing its attention distribution, ensuring the emphasis on both local and global features during extraction. This characteristic is important for tasks in the visual field. However, for neural network encoding, the situation

Submitted to 37th Conference on Neural Information Processing Systems (NeurIPS 2023). Do not distribute.

Table 1: Performance of searched architectures using different NAS algorithms in DARTS [7] space on CIFAR-10 [5]. † denotes using cutout [2] as data augmentation.

| Model | Parameters (M) | Top1 Acc (%) | No. of archs |
|---|---|---|---|
| VGG-19 [18] | 20.0 | 95.10 | - |
| DenseNet-BC [4] | 25.6 | 96.54 | - |
| Swin-S [9] | 50 | 94.17 | - |
| Nest-S [17] | 38 | 96.97 | - |
| Ransom search | 3.2 | 96.71 | - |
| NASNet-A† [19] | 3.3 | 97.35 | 20000 |
| AmoebaNet-A† [14] | 3.2 | 96.66 | 27000 |
| PNAS [6] | 3.2 | 96.59 | 1160 |
| NAONet [11] | 28.6 | 97.02 | 1000 |
| ENAS† [13] | 4.6 | 97.11 | - |
| DARTS† [7] | 3.4 | 97.24 | - |
| GATES† [12] | 4.1 | 97.42 | 800 |
| CTNAS† [1] | 3.6 | 97.41 | - |
| TNASP† [10] | 3.7 | 97.48 | 1000 |
| NAR-Former† [15] | 3.8 | 97.52 | 100 |
| NAR-Former V2† | 3.5 | 97.54 | 100 |

Table 2: Average cost for one sample of NAS-Bench-101. The inference latency was measured on a machine with GeForce RTX 3090 GPU. The batch size was set to 1.

| | Encode(ms) | Infer(ms) | Total(ms) |
|---|---|---|---|
| NAR-Former | 2.4784 | 17.4864 | 19.9648 |
| NAR-Former V2 | 2.3722 | 5.2276 | 7.5998 |

may be somewhat different. Due to the softmax, Eq. (5) focuses almost all attention on the current node while ignoring adjacent nodes, making it difficult to capture the topology of the neural network well. The Eq. (2) restricts attention to connected nodes by introducing the adjacency matrix. Considering that each neighboring node of the current node contributes to the representation of the network topology, we use linear attention to calculate attention and normalization without losing the information of neighboring nodes. The exponential function in the softmax function provides the nonlinear ability and guarantees the nonnegativity of elements. When softmax is replaced by linear normalization, we introduce the sigmoid activation function to achieve these two purposes.

# B More experiments

## B.1 Experiments on Darts

One important application of accuracy prediction is network architecture search (NAS). Here, we follow the NAS experiment of NAR-Former and evaluate our proposed model in the DARTS search space. To ensure fairness, we follow the experimental details adopted in NAR-Former [15].

Experimental results are listed in Table1. NAR-Former v2 retains the advantages of NAR-Former and also performs well in network architecture search. Thanks to its superior model performance, it only requires evaluating a small number of candidate networks to achieve excellent results in the search process. Based on NAR-Former v2, we obtained a model that has fewer parameters while having higher Top1 Accuracy compared with other predictor-based NAS methods.

In addition, as shown in Table 2, NAR-Former v2 achieves faster inference speed compared to NAR-Former as it has a more concise architecture.

## B.2 Experiments on NNLQP

In this part, we use Mean Absolute Percentage Error (MAPE) and Error Bound Accuracy (Acc($\delta$)) to measure the deviations between latency predictions and ground truths [8].

The MAPE is defined as:

$$\text{MAPE} = \frac{1}{n} \sum_{i=1}^{n} \frac{|y_i - y_i'|}{y_i} \times 100\%. \tag{7}$$

MAPE is a non-negative number and the smaller value means the more precision prediction.

The Acc($\delta$) can be calculated by:

$$\text{Acc}(\delta) = \frac{1}{n} \sum_{i=1}^{n} \text{cnt}\left(\frac{|y_i - y_i'|}{y_i} \leq \delta\right) \times 100\%, \tag{8}$$

where cnt($x$) is a counting function, with a value of 1 when condition $x$ is satisfied. The larger the Acc($\delta$), the better the prediction performance.

## B.3 Experiments on NAS-Bench-201

For a more comprehensive comparison with baseline [15] in terms of accuracy prediction, we added the 10% error bound accuracy metric (ACC (10%)), which reflects the accuracy of predicting exact values, and the standard deviation indicator, which reflects the stability of the method, respectively. The results are shown in Tab. 3. The results of ACC (10%) show that our NAR-Former V2 outperforms NAR-Former [15] in the prediction of exact values, in addition to the relative ordering, which also reflects that our method learns a more reasonable representation. Comparing the standard deviations of the two methods shows that our NAR-Former V2 is more stable than NAR-Former [15].

Table 3: More metrics results for accuracy prediction on NAS-Bench-201 [3]

| Train/Test | Kendall's Tau | | ACC (10%) | |
|---|---|---|---|---|
| | 424/all | 1563/all | 424/all | 1563/all |
| NAR-Former [15] | 0.8631±0.0080 | 0.8967±0.0029 | 96.77 | 99.32 |
| NAR-Former V2 | 0.8735±0.0026 | 0.8875±0.0013 | 99.45 | 99.50 |

In this paper, we show the results of applying our method to predict accuracy and latency, two of the most frequently considered attributes in network design and deployment. Our approach can also be easily used to predict other attributes of neural networks, simply by changing labels and adjusting hyperparameters as needed. We use our NAR-Former V2 with the same hyperparameters as the experiments on the NASBench family to predict the testing loss. This experiment is conducted on NAS-Bench-201. 5% of the whole data is used as the training set and another 200 samples are used for validation. The results are shown in Tab. 4. Kendall's Tau is 0.851, indicating that our model is capable of predicting the relative ordering of the testing loss for the entire dataset (15625 samples) with high correlation. The value of the other three metrics MAPE, ACC(10%), and ACC(5%) demonstrate the high accuracy of our method in predicting the actual values of the testing loss.

## C Implementation details

### C.1 Encoding details

For accuracy prediction, the length of operation encoding and position encoding is both 64 and the total encoding length of each node is 128. For latency prediction, the length of operation type encoding is 32, and the length of encoding of each attribute (e.g. kernel size, number of groups, the height of tensor, the width of tensor, and so on) is 10. There are a total of 12 attributes for each node, including 8 parameters related to the model definition and 4 attributes that describe the shape of node output tensors. Therefore, in latency prediction, the length of a single node of the initial

Table 4: Testing loss prediction on NAS-Bench-201 [3]

| Metrics | Kendall's Tau↑ | ACC(10%)↑ | ACC(5%)↑ | MAPE↓ |
|---|---|---|---|---|
| NAR-Former V2 | 0.851 | 96.01 | 77.70 | 3.48 |

encoding is 152. The total encoding length of the four static attributes (batch size, memory access, parameter quantity, and FLOPs) is 40, which, together with the output of the transformer, is used for the final latency prediction.

## C.2 Model details

The output dimension of each Transformer block is 512. The ratio of the hidden dimension to the input dimension in the grouped feed-forward networks of the Transformer block is fixed to 1:4.

## C.3 Training details

### C.3.1 Latency prediction on NNLQP

We follow the NNLP[8] training setup to train each model for our latency prediction experiments. That is, the batch size is 16 and the number of epochs is 50.

### C.3.2 Accuracy prediction on NAS-Bench-101

The NAS-Bench-101[16] repeated the training and evaluation of all architecture on CIFAR-10 for three times. There are accuracies after training with 4 different epochs: 4, 12, 36, and 108. The accuracies after training with 108 epochs are adopted in our experiments. We use the average validation accuracy of each architecture as the training target and the average testing accuracy to evaluate the testing performance of the trained predictor.

Following the NAR-Former [15], we use the information flow consistency augmentation. The weight of MSE loss, SR_loss, and AC_loss is 1, 0.1 and 0.5, respectively. We determined to train our predictor for 3000 epochs based on the convergence of the learning curve on the training set.

### C.3.3 Accuracy prediction on NAS-Bench-201

The NAS-Bench-201[3] trained and evaluated each architecture on three datasets: CIFAR-10, CIFAR-100, and ImageNet-16-120. In our experiments, the accuracy of each architecture on CIFAR-10 is used. The validation accuracy and testing accuracy of each architecture are used for building training ground truth and testing ground truth, respectively.

The setting of the loss function is similar to that of NAS-Bench-101. The predictor is trained for 1000 epochs in this part.