# OpenReview forum: "NAR-Former V2: Rethinking Transformer for Universal Neural Network Representation Learning"
_NeurIPS.cc/2023/Conference — NeurIPS 2023 poster_

### Official Review · Reviewer_sJnp · 2023-07-01

**Soundness:** 3 good
**Presentation:** 3 good
**Contribution:** 3 good
**Rating:** 5
**Confidence:** 4

**Summary:**

This paper studies how to model and learn the representations of neural networks themselves. Inspired by Transformer and GNN, NAR-Former V2 is proposed which is a modified Transformer-based universal neural network representation learning model. Extensive experiments demonstrate its effectiveness.

**Strengths:**

- This paper is well-motivated. It is reasonable to incorporate graph-specific properties into the vanilla Transformer. A new graph-aided attention-based Transformer block is proposed. The novelty of this paper is good.
- Compared to the baseline NAR-Former, NAR-Former V2 improves latency prediction performance significantly. And the authors conduct experiments on the recently released benchmark. Hence, NAR-Former V2 can be considered as a new baseline in this field.
- The whole paper enjoys well writing and good organization.

**Weaknesses:**

- The latency of the proposed NAR-Former V2 is not considered. It is better to make a comparison of the model latency.
- In lines 331-338, please report the specific number when comparing NAR-Former V2 with NAR-Former.
- In Tab. 2, NAR-Former V2 suffers from poor performance on AlexNet and VGG. Both of them have no residual connections. Why does NAR-Former V2 perform badly on these networks?

**Questions:**

Please see weaknesses.

**Limitations:**

The authors have adequately addressed the limitations

---

> ### Author Rebuttal · Authors · 2023-08-09
>
> Thank you for your positive comments. The responses to your questions is as follows.
>
> >1. The latency of the proposed NAR-Former V2.
>
> The latency of the proposed NAR-Former V2 is presented in Table 3 of the supplementary.  More latency comparison experiments are provided as follows:
> |               |                 |             |  424/100      |  424/all     |  4236/all      |
> |---------------|-----------------|-------------|---------------|---------------|---------------|
> |               | Structure       | Latency(ms) | Kendall's Tau | Kendall's Tau | Kendall's Tau |
> | NNLP          | graph-based     |   1.4625      | 0.497         | 0.484         | 0.563         |
> | TNASP         | attention-based | 4.6623      | 0.754         | 0.722         | 0.820         |
> | NAR-Former    | attention-based | 17.4864     | 0.801         | 0.765         | 0.871         |
> | NAR-Former V2 | GraphAttn-based | 5.2276      | 0.802         | 0.773         | 0.861         |
>
> The "412/100" notation signifies that there are 424 training samples and 100 test samples. As indicated in the table, the NNLP model, based solely on graphs, demonstrates the fastest inference speed due to its streamlined structure and computations. However, it faces challenges in accurately predicting outcomes. While the NAR-Former achieves comparable Kendall's tau results to our approach, its notably longer inference time hampers its efficiency. In contrast, in comparison to the exclusively attention-based TNASP model, our method maintains a similar latency level  while significantly enhancing prediction accuracy.
>
> >2. Report the specific number in lines 331-338
>
> Thank you for your suggestion. We have revised lines 331-338 as follows based on your suggestion: Regarding the average Kendall's Tau under different training settings, our method is on par with NAR-Former with 0.812 on nas-bench-101 and slightly higher than NAR-former by 0.6\% on NAS-Bench-201. Therefore, compared to NAR-Former, NAR-Former V2 achieves comparable accuracy prediction performance. In latency prediction experiments, our proposed model exhibits a clear advantage over NNLP, and outperforms NAR-Former by a significant margin. In summary, by incorporating the strengths of GNN, the universal representation learning framework NAR-Former V2 is significantly enhanced. NAR-Former V2 addresses the shortcomings of NAR-Former, which was overly sensitive when handling complete network structures, while still retaining the outstanding performance of NAR-Former when handling cell-structured networks.
>
> >3.  Poor performance on AlexNet and VGG.
>
> We conjecture that this is mainly due to the fact that, compared to the other networks, the gap in data distribution between the training and test set is larger when using AlexNet and VGG as test sets, making prediction more difficult. This large difference in data distribution is reflected in two main areas. One aspect lies in the structure of the neural network. With the exception of AlexNet and VGG, all other networks are constructed by stacking cells or modules. This leads to the problem that the structure of the network to be encoded at the testing phase will be far from the vast majority of networks seen in the training stage. On the other hand, the gap between the depth of these two networks and the average network depth during training is the largest, and this difference in network size will also further enhance the difficulty of encoding and prediction during testing.

---

### Official Review · Reviewer_kdkx · 2023-07-04

**Soundness:** 3 good
**Presentation:** 2 fair
**Contribution:** 3 good
**Rating:** 5
**Confidence:** 2

**Summary:**

This paper proposes to design graph-aided attention Transformer blocks that can be used to model and learn the representation of neural networks. Specifically, one can leverage this kind of representation learning framework to provide more accurate latency and accuracy predictions of given deep neural networks.

The proposed method surpasses the GNN-based NNLP baseline in latency estimation on the NNLQP dataset, and improves accuracy prediction on the NASBench101 and NASBench201 datasets.

**Strengths:**

* Leveraging graph topology to help the attention networks seems new to me. And the target problem of how to predict latency or accuracy is of great significance to the NAS and representation learning community.

* Results indeed show better predictions for latency and accuracy as compared to baselines.

**Weaknesses:**

* The author does not provide a context for what "NAR" refers to. Also, the writer assumes all the audience is familiar with NAR-Fromer, which is not the case. Most audiences are still new to this field so better to make it more accessible to a general audience.

* The paper organization could be better, it starts with quite some details telling us what is Graph-aided attentions. But why should we care about it? I would suggest moving the Fig. 2 to the front, so the readers know what is going on and what module exactly are we talking about. Without this big picture, one can easily get lost and lose interest in this work.

* How is the proposed representation learning methods surpass other network selection algorithms, like general NAS algorithms as they are also designed with the philosophy of predicting accuracy given networks? In terms of both prediction accuracy and computational costs.

* I would like to know the efficiency of the proposed techniques as compared to pure attention-based or pure graph-based variants.

**Questions:**

See weaknesses.

**Limitations:**

N/A. Not provided in the main context.

---

> ### Author Rebuttal · Authors · 2023-08-09
>
> Many thanks for your review and detailed sugession. Next, we will endeavor to address your concerns point by point.
>
> > 1. Provide a context for what "NAR" refers to.
>
> Thank you very much for your suggestions; this was an oversight on our part. We should consider more for the audiences who are new to this field.  In the introduction part, the definition of the task and a brief introduction has been added in the revised version.
>
> > 2. Reorganizing the structure of the article
>
> Thank you again for your suggestions for the details of our article organization. We have given this issue serious consideration. We agree with you that placing Figure 2 up front might give the reader a quicker understanding and a clearer idea of the role of the modules we are describing (graph-aided attention) in the overall model that corresponds to the task. New origination has been adopted in the revised version.
>
> >3. Comparison with NAS algorithms.
>
> Table 1 in the supplementary material presents a comparison of NAS experiments. The experimental results demonstrate that the proposed Nar-Former v2 assists NAS algorithms in discovering superior architectures, characterized by higher accuracy and fewer parameters compared to architectures identified by other NAS algorithms.
>
> >4.  The efficiency of the proposed techniques as compared to pure attention-based or pure graph-based variants.
>
> We complement the latency information for different backbone-based methods on NAS-Bench-101 in the table below:
>
> |               |                 |             |  424/100      |  424/all     |  4236/all      |
> |---------------|-----------------|-------------|---------------|---------------|---------------|
> |               | Structure       | Latency(ms) | Kendall's Tau | Kendall's Tau | Kendall's Tau |
> | NNLP          | graph-based     | 1.4625            | 0.497         | 0.484         | 0.563         |
> | TNASP         | attention-based | 4.6623      | 0.754         | 0.722         | 0.820         |
> | NAR-Former    | attention-based | 17.4864     | 0.801         | 0.765         | 0.871         |
> | NAR-Former V2 | GraphAttn-based | 5.2276      | 0.802         | 0.773         | 0.861         |
>
> The "412/100" indicates that the number of training and test samples are 424 and 100 respectively. As can be seen from the table, the pure graph-based approach NNLP has the fastest inference speed due to the simplicity of the structure and computation. However, it suffers from the very poor performance of accuracy prediction. Although NAR-Former can achieve kendall's tau results that are as advanced as our method, its quite longer inference time makes it less efficient. Compared to the pure attention-based TNASP model, we have the same level of latency (only 0.6ms more), but achieve a significant improvement in prediction accuracy (4.7% on average).

---

> > ### Comment · Reviewer_kdkx · 2023-08-16
> > **Response to the rebuttal**
> >
> > Thank you for providing the detailed rebuttal. Many revisions were promised but I feel hard to find them in the paper without any highlights.
> >
> > And after reading other reviewers' comments. I agree that the performance improvements seem marginal. And the writing style seems vague, i.e., a lot of intuition without grounding data to support or without clear explanation. Also, there are typos in the rebuttal text, e.g., "origination" should be "organization"?
> >
> > Considering pros and cons, I will keep the current rating with a low confidence.

---

> > > ### Author Response · Authors · 2023-08-18
> > > **Author response to Reviewer kdkx**
> > >
> > > We appreciate your feedback. Here are our responses.
> > >
> > > > Being hard to find the promised revisions in the paper without any highlights.
> > >
> > > Thank you and the other reviewers for the many valuable suggestions that led us to revise the paper into a more readable and rigorous version. For many of the questions in the rebuttal, we have corresponding short versions of the answers or explanations in the submission. However, thanks to the comments of the reviewers, we were inspired to elaborate on these issues in detail as demonstrated in the rebuttal. For example, the question about the choice of baseline mentioned by reviewer L4c1 is briefly answered in lines 261 to 263 of the submission. Based on the inspiration from the reviewer's comment, we demonstrate the rationality of our choice of baseline through additional experiments. Therefore, many of the suggestions given by all the reviewers helped us to revise our paper to be more readable and rigorous.
> > > Below is a portion of our revised paper:
> > >
> > > (1) Line 47:
> > > ... As shown in the latency prediction experiment in NAR-Former **(Transformer-based neural architecture representation learning framework)**...
> > >
> > > (2) We  will make the overall architecture diagram (original Figure 2) the first diagram, and add the following blurb at the beginning of Section 3.1:
> > >
> > > **As similar to our final framework in Figure. 1, existing neural network representation learning and attribute prediction methods consist of three main phases: neural network encoding, backbone-based representation learning, and attributes predicting using prediction heads. In this subsection, we will mainly focus on the backbone-based representation learning part to introduce our motivation.**
> > >
> > > (3)Add other experiments, details, etc. mentioned by other reviewers
> > >
> > > Unfortunately, at the moment, the system does not allow uploading our updated papers. But all the changes mentioned in the rebuttal have been already added in the revised version.
> > >
> > > > The performance improvements seem marginal.
> > >
> > > For the performance improvements of our proposed NAR-Former V2, there is evidence that our method is an improved version of NAR-Former and achieves SOTA in encoding both small cell components and complete deep neural networks. On the one hand, as can be seen from Tables 1 and 2 in the main text and the standard deviation comparisons in the table below, we more stably achieve accuracy prediction results comparable to the SOTA NAR-Former. More importantly, the results presented in Table 3 of the supplementary materials demonstrate that we have solved the problem of NAR-Former's poor generalization ability when encoding complete neural networks. On the other hand, the results in Tables 1 and 2 in the main text show that we achieve a considerable improvement compared to the SOTA of the complete deep neural network encoding.
> > >
> > > Standard deviation comparison on NAS-Bench-201:
> > >
> > > |               | 781/all                | 1563/all               |
> > > |---------------|------------------------|------------------------|
> > > | NAR-Former    | 0.8631$\pm$0.0080 | 0.8967$\pm$0.0029 |
> > > | NAR-Former V2 | 0.8735$\pm$0.0026 | 0.8875$\pm$0.0013 |
> > >
> > > > The writing style seems vague.
> > >
> > > For the statements and designs in the paper, we have verified them by corresponding experiments. We apologize for the typos and the absence of some detailed explanations, which will be amended and added in the final version based on the reviewers' comments.
> > >
> > > We thank you again for your response. If you have any further questions, suggestions, or concerns, please let us know. We believe that this will further help us to improve our paper!

---

### Official Review · Reviewer_L4c1 · 2023-07-05

**Soundness:** 3 good
**Presentation:** 3 good
**Contribution:** 2 fair
**Rating:** 6
**Confidence:** 3

**Summary:**

This submission builds on previous work and proposes NAR-Former V2, an updated transformer architecture to learn representations of neural network architectures. NAR-Former V2 targets NAS tasks, predicting model properties like performance or latency, to improve NAS runtime and cost. NAR-Former V2 makes use of a proposed graph-aided attention block, which uses the network's adjacency matrix to condition the attention matrix. The proposed NAR-Former V2 is empirically evaluated on NASBench101 and NASBench201 where it achieves comparable performance to state-of-the art models.



**Strengths:**

The submission is generally well written. It identifies a specific problem,latency and accuracy prediction for NAS, for which separate approaches have so far been suitable. The proposed method bridges that gap as a unified property predictor that covers both aspects.

The idea of using network connectivity by conditioning on the adjacency matrix of the network's graph is well motivated and convincing.
The empirical evaluation follows previous work and is thorough. The method achieves its goal of closing the performance gap between NNLP and NAR Former on latency and accuracy prediction.



**Weaknesses:**

While I find the submission generally of high quality, there are a few weaknesses.

While it may not have been the goal for this work, there are only small improvements over NAR-Formver v1, particularly on NASbench family. On these results (Table 3 and 4) there are also only average performance values reported. I'd ask the authors to add standard deviations or similar metrics, since the performance differences are very small.
Further, the submission shows different baselines for different experiments. While some numbers are mentioned in the text, I would find it more convincing to have the two main baselines the authors identified (NNLP and NAR-Former) in all results.

Also, the authors propose their method for universal neural network representation learning. However, they only show results on latency prediction and accuracy, which are important but relatively coarse labels. NasBench201 at least contains more detailed information on each model, which may be used to assess the quality of the representation on a more fine-grained level.

Lastly, there is an entire field of work that attempts to learn representations of neural networks, for example to predict model properties. While many of them use the actual weights, some use graph-based features. E.g.,
- You et al, 2020: Graph Structure of Neural Networks
- Corneanu et al, 2020: Computing the Testing Error Without a Testing Set.
They may merit discussion in the related work and may be used as different baselines.

**Questions:**

Question 1: Have you encountered any kind of ‘receptive field’ effect? Conditioning on the adjacency matrix should lead to several hops being necessary to aggregate information globally, unless I'm mistaken? What is your intuition into why that doesn’t seem to be an issue.

Question 2: Have you investigated other ways of encoding graph connectivity in transformers, e.g. like Kim et al, 2022: Pure Transformers are Powerful Graph Learners?



**Limitations:**

Not discussed explicitly.

---

> ### Author Rebuttal · Authors · 2023-08-09
>
> We appreciate the many detailed and insightful questions you have raised. Below are our responses.
>
> > 1. add standard deviations or other metrics in Table 3 and Table 4
>
> Thanks for your suggestion. Indeed, this is a good way to evaluate two methods. Compared with Nar-Former, Nar-Former V2 exhibits greater stability. For the accuracy prediction experiments on NAS-Bench-201, when employing a more stringent error bound, the advantages of Nar-Former v2 can also be demonstrated. We report more results using the 10% error bound accuracy metric (ACC(10%)) for further comparison with NAR-Former. The results are as follow:
>
> | Train/test    | 781/all | 1563/all |
> |---------------|---------|----------|
> | NAR-Former    | 96.77   | 99.32    |
> | NAR-Former V2 | 99.45   | 99.50    |
>
> These results will be included in the revised version.
>
> > 2. Utilize Nar-Former and NNLP as baseline models across all experiments.
>
> Nar-Former has already been validated in the accuracy prediction experiments and latency prediction experiments (as presented in the supplementary material). Here, we provide additional experiments regarding NNLP's performance in accuracy prediction. The experimental results are as follows:
> | Dataset    | 101     | 101     | 101      | 201     | 201     |
> |------------|---------|---------|----------|---------|---------|
> | Trian/Test | 424/100 | 424/all | 4236/all | 781/all | 781/all |
> | KT of NNLP | 0.497   | 0.484   | 0.563    | 0.675   | 0.703   |
>
> The table ablove shows the accuracy prediction results of NNLP on the NAS-Bench-101 and NAS-Bench-201, which are significantly worse than the performance of NAR-Former.  As described in our introduction, Transformer models currently hold an advantage in accuracy prediction tasks.  Therefore, the NAR-Former is utilized as the baseline for accuracy prediction. For latency prediction, the results in Table 1, Table 2 of the main text and Table 3 of supplementary material demonstrate that NNLP is the SOTA model, which is then adopted as the baseline for this task.
>
> >3. Predicting more attributes with the proposed NarFormer-V2 on NasBench201
>
> In this paper, we show the results of applying our method to predict accuracy and latency, two of the most frequently considered attributes in network design and deployment. Actually, our approach can also be easily used for other attribute predictions of neural networks, simply by changing labels and adjusting hyper parameters as needed. We use our NAR-Former V2 with the same hyper parameters as the experiments on NASBench family to predict the testing loss. The results are shown in the table below:
>
> | Metrics       | Kendall's Tau$\uparrow$ | ACC(10%)$\uparrow$ | ACC(5%)$\uparrow$ | MAPE$\downarrow$ |
> |---------------|---------------|----------|---------|------|
> | NAR-Former V2 | 0.851         | 96.01    | 77.70   | 3.48 |
>
> The Kendall's Tau is 0.851, indicating that our model is capable of predicting the relative ordering of the test loss for the entire dataset (15625 samples) with high correlation. The value of the other three metrics MAPE, ACC(10\%), and ACC(5\%) demonstrate the high accuracy of our method in predicting the actual values of loss for all networks in the dataset. In the revised version, we will incorporate additional experimental results.
>
> > 4. Some references
>
> Thanks for the reminders. We provide an introduction to some common and classical neural network representation learning methods in the first part of the related work section. Regrettably, maybe it is not comprehensive enough. We will add some discussions later according to your suggestion.
>
> > 5.The receptive field effect
>
> Yes, we have encountered some receptive field effect. Each layer of our graph-aided attention can only observe neighboring nodes. As the graph-aided attention-based transformer stack up, the receptive field gradually expands. Consistent with existing experience, when containing very few attention layers, e.g., only 1 layer, the receptive field is too small, leading to poor performance. However, in the task of this paper, there are some differences from the common sense that the larger the receptive field the better the result usually is. When the receptive field is too large, such as when using the vanilla global attention-based transformer model, the encoding of neural networks with depths up to hundreds of layers tends to unsatisfactory.
>
> > 6. Other ways of encoding graph connectivity in transformers
>
> While there are indeed some other works to encode graphical connectivity in the Transformer [1, 2, 3], we are different from them in terms of what each focus on, motivation, the details of design, etc.
>
> For example, the TokenGT [1] mainly proposes to build graph learners directly using pure transformers by treating all nodes and edges as tokens in order to exploit the advantages of transformer in conjunction with large-scale training. But instead, we focus on modifying the transformer using GNN's inductive bias to improve its generalization ability and performance on representation acquisition and attribute prediction of very deep neural networks. In order to learn the representation of graph-structured data, some methods also allow nodes to attends only to their local node neighbors in the attention computation of Transformers, but our motivation and goals are different from theirs [2].
>
> In addition, we are the first attempt to analyze the impact of graph connectivity properties and global connectivity properties in the transformer on the task of neural network encoding and representation learning. It can be observed from the extensive experiments in the main text, supplementary material and rebuttal PDF that our analysis is sound and our modifications are effective.
>
> **Reference**
>
> [1] Kim et al, 2022, Pure Transformers are Powerful Graph Learners.
>
> [2] Dwivedi et al, 2021, A Generalization of Transformer Networks to Graphs.
>
> [3] Ying et al, 2021, Do Transformers Really Perform Bad for Graph Representation?

---

> > ### Comment · Reviewer_L4c1 · 2023-08-13
> > **Reviewer response to rebuttal**
> >
> > I'd like to thank the authors for the extensive response, which is much appreciated.
> >
> > I'm not entirely sure how to understand the 'more stringent 10% error bound'. Is there any reason why the authors do not report mean +- std of the performance metric? I feel like this has become standard and is accessible to the reader, while other metrics are more obscure.
> >
> > The additional experiments improve the submission in my view, in particular the rank order of performance measured in kendall's tau. This seems to be a much more relevant and robust metric to make a choice between different architectures, and increases the evidence for the expressiveness of the method. I have therefore increased the score.

---

> > > ### Author Response · Authors · 2023-08-15
> > > **Author response to Reviewer L4c1**
> > >
> > > First of all, we really appreciate your positive comments and raising our score! Below are the responses to the questions you asked.
> > >
> > > > How to understand the 'more stringent 10% error bound'.
> > >
> > > A definition of error bound accuracy is presented in the supplementary material. In contrast to Kendall's Tau, which focuses on the correlation between the relative ordering of the predicted values and that of the true values, the assessment of the error bound accuracy is based on the exact error between the predicted value and the true value. Thus here we describe it as a more "stringent" metric.
> > >
> > > > Is there any reason why the authors do not report mean +- std of the performance metric?
> > >
> > > The results of the comparison of error bound accuracy show that our NAR-Former V2 outperforms NAR-Former in the prediction of exact values, in addition to the relative ordering, which also reflects that our method learns a more reasonable representation. We regret that we neglected to compare standard deviations. We performed 12 replicate experiments for each method, and calculate the mean and standard deviation of the Kendall's Tau values after removing the maximum and minimum. The results are shown in the following table:
> > >
> > > |               | 781/all                | 1563/all               |
> > > |---------------|------------------------|------------------------|
> > > | NAR-Former    | 0.8631$\pm$0.0080 | 0.8967$\pm$0.0029 |
> > > | NAR-Former V2 | 0.8735$\pm$0.0026 | 0.8875$\pm$0.0013 |
> > >
> > > Comparing the standard deviations of the two methods shows that our NAR-Former V2  is more stable than NAR-Former.
> > >
> > > Finally, thank you again for your feedback. If you have any further questions or suggestions, please let us know. We will be happy to answer them for you.

---

### Official Review · Reviewer_9nHL · 2023-07-18

**Soundness:** 2 fair
**Presentation:** 2 fair
**Contribution:** 2 fair
**Rating:** 5
**Confidence:** 4

**Summary:**

The paper proposes a transformer based neural network representation learning model that incorporates the inductive representation capabilities of graph neural networks into the transformer architecture to improve it's generalizability on unseen architectures. The NAR-Former framework, which the proposed model builds on, has poor generalization performance when the depth of the input architecture gets large(eg: in the hundreds). The authors argue that for a neural network representation model to process neural architectures in varying tasks, a combination of both the Transformer and GNN architectures is necessary. The resulting model, NAR-Former V2, is a combination of the self-attention mechanism of transformers which discovers pair-wise correlations between its inputs and the graph aggregation mechanism of graph neural networks which is capable of discovering topological structures in its inputs.

**Strengths:**

- The proposed method can handle cell-structured networks and can learn representations of entire networks extending the capabilites of NAR-Former.

- The method is extensively benchmarked on various tasks such as attribute and latency prediction as well as on varaious NAS benchmarks.

- The proposed method is scalable and can encode networks with hundreds of operations.

- The paper is easy to read and follow.

**Weaknesses:**

- There are many claims made in the paper which, perhaps made by intution, are not supported quantitatively (See questions).

- Comparison with NAR-Former is missing for some crucial results used to validate the proposed method which is a direct modification to NAR-Former.

**Questions:**

- Lines[176-178] How does the Type-aware Enhancement module actually 'enhance' the representation? What does enhance mean here? Was there a comparison with the representations of NAR-Former's representations? The usage of the term seems a bit loose and i'm not sure how to intepret this. If this is by some metric? Then it is best to provide this to the reader.

- In Tables 3 and 4 where NAR-Former is compared with NAR-FormerV2, there seems to be no difference between the two models when averaged over all the test samples. Specifically, averaging the results in Table 3 gives 0.82 for both NAR-Former and the proposed NAR-FormerV2. Similarly for Table 4, taking an average over all testing settings gives 0.875 and 0.881 for NAR-Former and NAR-FormerV2 respectively. Since the proposed method is supposed to be a direct enhancement of NAR-Former, do these results still justify the modifications made to the original model?

- Since the proposed model is a direct improvement on NAR-Former, why isn't there a comparison with NAR-Former in Tables 1 and 2?

- What is the parameter count of NAR-Former V2 compared to NAR-Former? While the latency measurements on an RTX 3090 GPU is provided in Table 2, the actual parameter count could be useful.

- In the ablation study (Table 5), what happens when:
    - Only GFFN is incorporated in the Transformer.
    - Only TA-Enhance is incorporated in the Transformer.
    - A combination of only GraphAtnn and TA-Enhance only.

  Table 5 suggests than combining all 3 leads to the best performance but the ablation ignores the combinations listed above. For instance it might be the case that combining either TA-Enhance or GFFN with the regular transformer based attention performs better than the combinations with the GraphAttn.

- In the Appendix(Lines 16-18), again claims that the inadequacy of Eq. 5 are directed at the usage of the softmax function with a description of why this is the case together with how Eq. 2 restricts attention to connected nodes only. Could the attention matrix and adjacency matrix be visualized to actually confirm if the models behave as the authors say? This would be a much better quantitative illustration of the benefits of the proposed model over regular Transformer based attention.

- [Minor] Typo on line 59.

**Limitations:**

No obvious limitations.

---

> ### Author Rebuttal · Authors · 2023-08-09
>
> We appreciate the valuable questions and suggestions you've given, and here are our responses.
>
> > 1. More explanation for the proposed Type-aware Enhancement module
>
> The ``Enhancement" in the Type-Aware Enhancement module refers to the use of intrinsic information of the input data to modify the hidden representations to make them more sensible and relevant to the structure of the input architecture. Since it is observed that the number of connections for each type of operation in a neural network contains certain patterns, e.g. the convolutional layer has only one input, while the addition operation consists of more than one input. Therefore, for each node, we take as input the number of nodes it is connected to, and obtain a modulation parameter to adjust the representation, in order to make them more relevant to the corresponding type of operation. The ablation experiments verified the rationality and validity of our design.
>
> > 2. The advantages in Tables 3 and 4 are not significant
>
> The main contribution of this paper is to address the poor generalization performance of NAR-Former on encoding complete deep neural networks while maintaining its state-of-the-art results on encoding cell structures. The results in Tables 3 and 4 indicate that our method can indeed maintain the outstanding performance of accuracy prediction on cell structures. However, more importantly, as shown in Table 3 of the appendix, we get a significant boost in encoding complete DNNs compared to NAR-Former and exceed the SOTA (NNLP) in latency prediction. Thus, overall, the method in this paper is an improved version of NAR-Former, which can reach the SOTA in two vastly different scenarios, that is, encoding cell and encoding complete DNNs.
>
> > 3. The direct comparison between Nar-Former v1 and Nar-Former v2
>
> We apologize for any confusion caused by placing the direct comparative experiments in the supplementary material due to time and space constraints. The reviewer rightfully pointed out the absence of these direct comparison experiments. In response, we acknowledge this oversight and are committed to rectifying it in the revised version by incorporating the direct comparative experiments into the main body of the paper.
>
> Tables 1 and 3 in the supplementary material provide a direct comparison of the performance of Nar-Former v1 and Nar-Former v2 in terms of network architecture search and latency prediction. The experimental results consistently demonstrate that Nar-Former v2 outperforms Nar-Former v1 in these aspects.
>
> > 4. Comparison of parameter counts
>
> A more comprehensive comparison with NAR-Former on NAS-bench-101 is as follows:
>
> |               | Params(M) | Latency(ms) |
> |---------------|-----------|-------------|
> | NAR-Former    | 4.7850    | 17.4876     |
> | NAR-Former V2 | 8.5825    | 5.2276      |
>
> Resulting from the fact that (1) our network has a larger width (512) than NAR-Former (192), and (2) in contrast to NAR-Former's transformer-based multi-stage fusion mechanism, we use the a simplified but  parameter-heavy fully-connected layer-based fusion scheme, our model in this setting has more parameters than NAR-Former. Even with more parameters, the simplicity of the computation of the fully-connected layer allows us to still achieve faster inference.
>
> > 5. Additional ablation experiments on 1) combining GFFN, TA-Enhance with Transformer 2) combining Graph Atnn and TA-Enhance.
>
> The results of the ablation experiments for the several scenarios you are concerned about (Row (1), (2), and (3)) are shown in the table below:
>
> | Row           | Structure   | GraphAttn | GFFN     | TA-Enhance | MAPE$\downarrow$  | ACC(10%)$\uparrow$ | ACC(5%)$\uparrow$ |
> |---------------|-------------|-----------|----------|------------|-------|----------|---------|
> | (a1)          | Transformer | &#10004;  | &#10004; | -          | 3.20  | 96.00    | 81.86   |
> | (1)           | Transformer | -         | &#10004; | -          | 10.24 | 60.62    | 36.91   |
> | (2)           | Transformer | -         | -        | &#10004;   | 9.16  | 64.96    | 40.05   |
> | (3)           | Transformer | &#10004;  | -        | &#10004;   | 3.09  | 96.15    | 82.76   |
> | NAR-Former V2 | Transformer | &#10004;  | &#10004; | &#10004;   | 3.07  | 96.41    | 82.71   |
>
>  As we can see from the Rows (a1) and (1), combining GFFN with the regular transformer (with global attention) performs much worse than the combinations with the GraphAttn.  Comparing the results in Rows (2) and (3) reveals that a similar conclusion holds for TA-Enhance. These again validates what we described in the paper that global attention may cause excessive sensitivity when encoding complete DNNs, learning too many non-general features from the training set data and thus showing poor generalization ability on the test set.
>
> > 6. Visualizing attention maps to validate our propositions
>
> For the visualization of attention map and adjacency matrix, please refer to figure of the PDF. Subplot (a) of Fig. 1 is the raw attention graph that has not yet been normalized. We analyze the 15th node as an example (line 15 marked by the red box). We can draw conclusions from the Fig. 1: (1) The feature of softmax makes it overly focused, with most of the other nodes, which include neighboring nodes, being ignored and given very similar weights. This can be seen from the color distribution of the heat map in the red box in subplot (c). (2)  This neglect and equal weighting leads to  having similar attention to different neighbors in the softmax GraphAttn, which will likely cause the model to degenerate into the GNN aggregation rule. The linear GraphAttn avoids this problem. This can be seen in subfigures (d) and (f).

---

> ### Author Response · Authors · 2023-08-21
> **Author Comment to Reviewer 9nHL**
>
> We sincerely thank you for the  thoughtful feedback and comments you've given. Your investment of time in reviewing our work is greatly valued.
>
> Should you have any further questions or require additional clarification on any points, please do not hesitate to ask.
>
> Considering the clarifications provided, we would be grateful if you could reconsider the score you've assigned, in light of the recent explanations.
>
> Your understanding and continued support are deeply appreciated.

---

> ### Comment · Area_Chair_gaYu · 2023-08-22
> **Please provide feedback**
>
> Dear Reviewer 9nHL,
>
> The official author-reviewer discussion period is over. However, it would be great if you could go over the authors' responses and the reviews from the others and provide feedback.
>
> Thanks,
> AC

---

### Author Rebuttal · Authors · 2023-08-09

We thank all reviewers for their valuable comments. Reviewers 9nHL,L4c1 and sJnp comment this paper is well written and easy to read. Reviewers L4c1, kdkx and sJnp agree with our motivation and comment the idea of using network connectivity by conditioning on the adjacency matrix of the network's graph is novel (new) or convincing.

Next, we will address the reviewers' questions point by point.

---

### Decision · Program_Chairs · 2023-09-21

**Decision:**

Accept (poster)

**Comment:**

This paper introduces NAR-Former V2, a universal neural network representation learning method capable of encoding any neural network architecture. It addresses the limitations of NAR-Former, particularly its poor generalization with larger networks. Specifically, the authors combine the self-attention mechanism of Transformers with the graph aggregation mechanism found in graph neural networks (GNNs). This fusion enables the Transformer to incorporate the inductive learning capabilities of GNNs, allowing it to generalize effectively to previously unseen architectures. Experimental validation of the proposed method on multiple benchmark datasets for attribute and latency prediction tasks demonstrates its effectiveness.

Initially, the paper received borderline ratings, with reviewers leaning slightly toward rejection. Reviewers generally appreciated the paper's well-written nature, clear motivations, and the novel idea of integrating graph representation learning into Transformers. The extensive experimental validation also left them convinced that the proposed method could be practical.

However, concerns were raised about several unsubstantiated claims lacking quantitative results. Reviewers also expressed worry about the modest improvements of the proposed method over NAR-Former and the absence of a comparison between the two in some experimental settings. Additionally, the method's subpar performance on certain architectures (e.g., AlexNet and VGG) raised concerns. There was also a lack of discussion on related works and comparisons with them.

During the discussion period, many of these concerns were addressed through clarifications and additional experimental results provided by the authors. This led to a consensus among the reviewers that the paper's strengths outweigh its weaknesses, despite it still remaining on the borderline.